# BUNDLEFLOW: Deep Menus for Combinatorial Auctions by Diffusion-Based Optimization

**Tonghan Wang**
Harvard University
twang1@g.harvard.edu

**Yanchen Jiang**
Harvard University
yanchen_jiang@g.harvard.edu

**David C. Parkes**
Harvard University
parkes@eecs.harvard.edu

## Abstract

Differentiable economics—the use of deep learning for auction design—has driven progress in multi-item auction design with additive and unit-demand valuations. However, there has been little progress for combinatorial auctions (CAs), even in the simplest and yet important single bidder case, due to exponential growth of the bundle space with the number of items. We address this challenge by introducing a deep network architecture for a menu-based CA, which supports the first dominant-strategy incentive compatible (DSIC), revenue-optimizing single-bidder CA. Our idea is to generate a bundle distribution through an ordinary differential equation (ODE) applied to a tractable initial distribution. Our method, BUNDLEFLOW, learns suitable ODE-based transforms, one for each menu element, to optimize expected revenue. BUNDLEFLOW achieves up to $2.23\times$ higher revenue than baselines on standard CA testbeds and scales up to 500 items. Compared with other menu-learning baselines, BUNDLEFLOW also reduces training iterations by $3.6-9.5\times$ and cuts training time by about 80% in settings with 50 and 100 items.

## 1 Introduction

When selling multiple items simultaneously, bidders often have complex valuations that exhibit synergies among items. For instance, some items may act as complements, with their collective value to a bidder exceed the sum of their individual values. *Combinatorial auctions* (CAs) support such valuations by allowing bids on bundles of items. Recognized as early as 1922 [54] and formally defined in 1982 [43], CAs have found a lot of application, from allocating airport runways to auctioning spectrum licenses [9, 41], efforts whose far-reaching impact underpinned the 2020 Nobel Prize in economics [46].

While multi-bidder CAs represent the most general framework, many real-world settings are single-bidder CAs: a streaming platform curating a collection of movies for a viewer (e.g., Prime Video UK publicly listed 65+ channels on March 2, 2022 [1]), a cloud provider packaging compute features (hundreds of AWS EC2 instance types with CPU/GPU, memory, bandwidth, and storage options [2]), an integrated facilities vendor offering configurable service lines (custodial, HVAC, security, pest control, energy, etc.) across multiple sites [4]. This problem has also been studied as the so-called "FedEx problem" [15]. The single-bidder CA model is also applicable in settings with multiple bidders as long as supply constraints apply independently to each bidders; e.g., with information goods, or services that can be replicated and sold any number of times.

Despite their prominence, designing revenue optimal CAs remains fundamentally challenging. As with auctions for additive or unit-demand valuations, it is typical to seek mechanisms that are (1)

39th Conference on Neural Information Processing Systems (NeurIPS 2025).

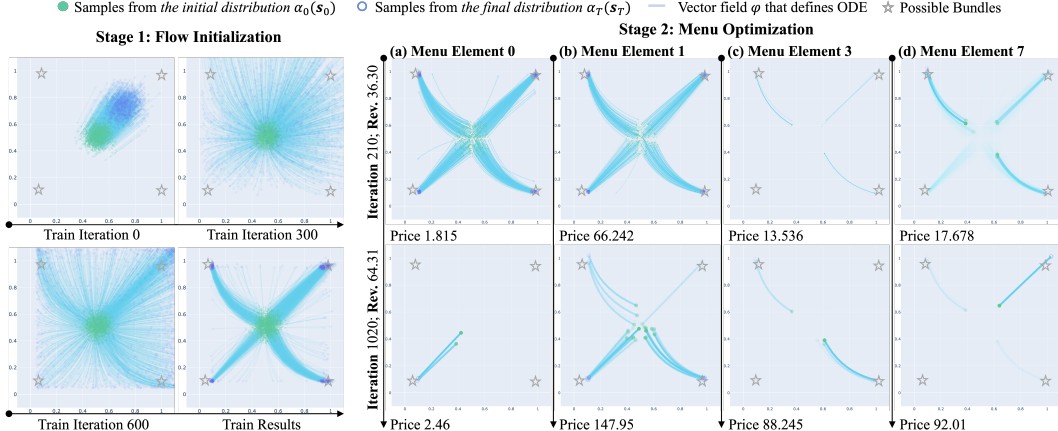

Figure 1: Demonstration of the BUNDLEFLOW method. The x- and y-axes represent the bundle variables for two specific items: $x = 1$ and $y = 1$ correspond to item A and B, respectively, being in the bundle. An ODE $d\boldsymbol{s}_t = \varphi(t, \boldsymbol{s}_t)dt$ (blue curves) generates bundle distributions $\alpha_T(\boldsymbol{s}_T)$ (blue dots) from simple initial distributions $\alpha_0(\boldsymbol{s}_0)$ (green dots). The opacity represents probability density. **Left**. Stage 1: Fix $\alpha_0(\boldsymbol{s}_0)$ and train the vector field, so the final distribution has all feasible bundles as its support. **Right**. Stage 2: Fix the vector field and update initial distributions of menu elements to manipulate bundle distributions. We present snapshots of four menu elements (organized in columns) at different training iterations (organized in rows), showing the bundle distribution and price for each element, along with the test-time auctioneer revenue from the entire menu at the corresponding iteration (see Sec. 4).

*dominant-strategy incentive compatible* (DSIC, also strategy-proof), ensuring that bidders benefit most by reporting their true values, and (2) revenue-maximizing from the auctioneer's perspective. However, even the seemingly simpler single-bidder combinatorial setting—a foundational building block for multi-bidder scenarios—lacks a comprehensive theoretical characterization.

In recent years, researchers have explored the use of deep learning techniques for optimal auction and mechanism design, commonly referred to as *differentiable economics* [13]. In particular, deep learning together with menu-based methods show promise. This learns a menu of options for a bidder, guaranteeing strategy-proofness provided the menu remains *self-bid independent* and *agent-optimizing* [20]. For a CA, each option in a menu corresponds to a bundle of items (or a distribution on bundles) and a price. Following *RochetNet* [13], various methods have been developed for the single-bidder, non-combinatorial setting [10, 11, 13, 48], demonstrating the ability to discover auctions that are provably optimal.

However, differentiable economics has made little headway on the problem of optimal CA design due to the fundamental challenge of handling a number of bundles exponential in the number of items. Dütting et al. [13] deal with two items and do not guarantee exact DSIC. Duan et al. [12] restrict their attention to the class of the *virtual valuation combinatorial auction* (VVCA), which is not fully general, and only scale to 10 items. Ravindranath et al. [44] consider a sequential CA setting, which is generally suboptimal when items can be simultaneously auctioned. No previous work provides a path towards DSIC and flexible, *i.e.*, fully general, mechanisms for tens or hundreds of items, and novel methodology is needed to make progress.

To understand the challenge of optimal CA design we compare menus for simpler valuations with menus for combinatorial valuations. For a bidder whose value is additive on items, it suffices to specify in a menu element the *item-wise allocation probabilities*, such as $[0.3, 0.6]$ for two items, along with a price. However, an item-wise allocation is ambiguous for a bidder with a more general valuation function. For example, $[0.3, 0.6]$ could be $0.3[1, 0] + 0.6[0, 1]$ or $0.3[1, 1] + 0.3[0, 1]$, with the item-wise allocation decomposed as different *bundle-wise allocations*, leading to distinct valuations. One could predefine how to interpret a marginal item-wise allocation to avoid ambiguity, for example adopting product distribution semantics. But such interpretations sacrifice flexibility–many bundle-wise allocations cannot be represented; e.g., $0.5[1, 1, 0] + 0.5[0, 0, 1]$ cannot be represented as a product distribution. One way to address this is for a menu element to directly specify an allocation probability for each possible bundle. However, the number of bundles grows exponentially with the number of items, making an explicit representation intractable.

We solve this problem, developing a menu-based, deep learning method for DSIC and fully general, single-bidder CAs that scales to as many as 500 items. The core idea is to avoid the need to directly specify a probability for every possible bundle. Instead, we represent a distribution on bundles for a menu element by a tractable and low-dimensional *initial distribution*, $\alpha_0(s_0)$, on *initial bundle variables*, $s_0 \in \mathbb{R}^m$, and an ordinary differential equation (ODE): $ds_t = \varphi(t, s_t)dt$. The ODE operates on *bundle variables*, $s_t \in \mathbb{R}^m$, through the *vector field* $\varphi(t, \cdot)$. Here $m$ is the number of items and $t \in [0, T]$ is the *ODE time*. A feasible bundle corresponds to a bundle variable where the entries are all 0s or 1s.

Formally, $s_t$ is generated from a sample $s_0$ by applying the ODE: $s_t(s_0) = s_0 + \int_0^t \varphi(\tau, s_\tau)d\tau$. We omit the dependence on $s_0$ and write $s_t$ for simplicity. In this way, the ODE transforms the initial distribution $\alpha_0(s_0)$ to a final distribution $\alpha_T(s_T)$ at time $T$. By designing the functional form of the vector field and using the *Liouville equation* [36], we get a tractable closed-form expression of $\alpha_T(s_T)$ that is fully differentiable and supports gradient-based training. Based on this, our method proceeds in two steps: (1) train the vector field so that the support of the final distribution $\alpha_T(s_T)$ corresponds to feasible bundles; (2) fix this trained vector field $\varphi(t, \cdot)$, and backpropagate the revenue-maximizing loss through the flow to update the initial distribution $\alpha_0(s_0)$ (which updates the final distribution over bundles) and the price for each menu element. A demonstration of these two steps is shown in Fig. 1.

This method draws inspiration from *diffusion models* [22, 32, 50, 51, 60] used in generative AI and especially *continuous normalizing flow* [8, 37, 38]. The technical novelty is to extend generative models to solve an optimization problem, seeking a bundle distribution (and price) for each menu element that optimizes expected revenue. In particular, there is no known target distribution in our work. By contrast, in classic generative AI tasks, the target distribution is known and observed as the data distribution in large-scale pre-training datasets of natural images [14, 45], language [39], and videos [53]. We call our method BUNDLEFLOW to emphasize the core idea of using continuous normalizing flow to optimize bundle distributions.

We evaluate BUNDLEFLOW using single-bidder instantiations of CATS [35], a widely adopted CA testbed. Experimental results demonstrate that our method consistently and significantly outperforms all baselines across all benchmark settings and scales to auctions involving up to 500 items. For auctions with 50 to 150 items, BUNDLEFLOW achieves $1.11-2.23\times$ higher revenue. Moreover, this substantially improved revenue does not compromise training efficiency. On the contrary, compared to the baseline RochetNet that also learns allocations in menu elements, our method typically requires $3.6-9.5\times$ fewer training iterations and reduces training time by about 80% in settings with 50 or 100 items.

We also make a critical observation when each menu element is forced to deterministically assign a single bundle. The revenue in this case drops sharply—by as much as 52.7%—bringing performance down to baseline levels. This underscores the necessity of supporting randomized bundle distributions in differentiable economics for CAs, a feature that is unique to our method. We discuss a direction to extend this work to multi-bidder CA design in Appx. B.

## 2 Preliminaries

**Combinatorial Auction**. We consider CAs with a single bidder and $m$ items, $M = \{1, \ldots, m\}$. The bidder has a *valuation function*, $v : 2^M \to \mathbb{R}_{\geq 0}$. $v$ is drawn independently from a distribution $F$ defined on the space of possible valuation functions $V$, determining how valuable each bundle $S \in 2^M$ is for the bidder. We consider bounded valuation functions: $v(S) \in [0, v_{\max}], S \subset 2^M$, with $v_{\max} > 0$, normalized so that $v(\varnothing) = 0$. The auctioneer knows distribution $F$ but not the valuation $v$. The bidder reports their valuation function, perhaps untruthfully, as their *bid (function)*, $b \in V$.

We seek an auction $(g, p)$ that maximizes expected revenue. Here, $g : V \to \mathcal{X}$ is the *allocation rule*, where $\mathcal{X}$ is the space of feasible allocations (i.e., no item allocated more than once), so that $g(b) \subseteq M$ denotes the set of items (perhaps empty) allocated to the bidder at bid $b$. Also, $p : V \to \mathbb{R}_{\geq 0}$ is the *payment rule*, specifying the price associated with allocation $g(b)$. This is a challenging design problem that has not been solved computationally or analytically in prior work. The utility to the bidder with valuation function $v$ at bid $b$ is $u(v; b) = v(g(b)) - p(b)$, which is the standard model of quasi-linearity so that values are in effect quantified in monetary units, say dollars. In full generality,

the allocation and payment rules may be randomized, with the bidder assumed to be risk neutral and seeking to maximize their expected utility.

In a *dominant-strategy incentive compatible* (DSIC) or *strategy-proof (SP)* auction, the bidder's utility is maximized by bidding their true valuation $v$, whatever this valuation; i.e., $u(v; v) \geq u(v; b)$, for $\forall v \in V, \forall b \in V$. An auction is *individually rational* (IR) if the bidder receives a non-negative utility when participating and truthfully reporting: $u(v; v) \geq 0$, for $\forall v \in V$. Following the revelation principle, it is without loss of generality to focus on SP and IR auctions, as any auction that achieves a particular expected revenue in a dominant-strategy equilibrium can be transformed into an SP and IR auction with the same expected revenue. Optimal auction design therefore seeks to identify an SP and IR auction that maximizes expected revenue, i.e., $\mathbb{E}_{v \sim \boldsymbol{F}}[p(v)]$.

**Menu-Based CAs**. In a *menu-based auction*, allocation and payment rules are represented by a *menu*, $B$, consisting of $K \geq 1$ *menu elements*. We write $B = (B^{(1)}, \ldots, B^{(K)})$, and the $k$th *menu element*, $B^{(k)}$, specifies allocation probabilities on bundles, $\alpha^{(k)} : 2^M \to [0, 1]$, and a *price*, $\beta^{(k)} \in \mathbb{R}$. We allow randomization, where $\alpha^{(k)}(S) \in [0, 1]$ is the probability that bundle $S \in 2^M$ is assigned in element $k$. The menu $B$ corresponds to a *menu-based representation of an auction*. A bidder with bid $b$ is assigned the element from menu $B$ that maximizes their utility according to the reported valuation: $k^* \in \arg\max_k \sum_{S \in 2^M} \alpha^{(k)}(S)b(S) - \beta^{(k)}$. We denote this optimal element by $(\alpha^*(b), \beta^*(b))$. The use of menu-based representations for auction design is DSIC and without loss of generality [20]. The optimal auction design problem is to find a menu-based representation that maximizes expected revenue, i.e., $\mathbb{E}_{v \sim F}[\beta^*(v)]$. Menu-based architectures [13, 48] in the differentiable economics literature [10, 11, 16, 24, 29, 30, 42, 56, 61, 63] learn to generate such menus by neural networks but have not been applied to anything other than very small CA problems.

**Diffusion Models and Continuous Normalizing Flow**. Diffusion models perform a forward noising process in which noise is incrementally added to training data over multiple steps, gradually corrupting the original sample. A reverse diffusion process is then learned to iteratively remove noise, thereby reconstructing data from near-random initial states. A *deterministic reverse process* modeled by an *ordinary differential equation* (ODE) preserves the same marginal probability densities as a reverse process modeled by a stochastic differential equation (SDE) [51]. A *continuous normalizing flow* also learns to transform and manipulate distributions by an ODE, with a continuous normalizing flow transporting an input $\boldsymbol{x}_0 \in \mathbb{R}^\ell$ to $\boldsymbol{x}_t = \phi(t, \boldsymbol{x}_0)$ at timestep $t \in [0, T]$. Here, $\phi(t, \cdot) : \mathbb{R}^\ell \to \mathbb{R}^\ell$ is the *flow*, and is governed by the ODE,

$$\frac{d}{dt}\boldsymbol{x}_t = \varphi\left(t, \boldsymbol{x}_t\right), \tag{1}$$

where the vector field $\varphi : [0, T] \times \mathbb{R}^\ell \to \mathbb{R}^\ell$ specifies the rate of change of the state $\boldsymbol{x}$. The flow $\phi$ transforms an initial distribution $p_0(\boldsymbol{x})$ to a final distribution $p_T(\boldsymbol{x})$ at time $T$. Continuous normalizing flow methods [8] suggest to represent the vector field $\varphi$ with a neural network.

**Rectified Flow**. A bottleneck that restricts the use of continuous normalizing flow in large-scale problems is that the ODE (Eq. 1) is hard to solve when the vector field $\varphi$ is complex. The *rectified flow* [38] addresses this by encouraging the flow to follow the linear path:

$$\min_{\varphi} \int_0^T \mathbb{E}_{\boldsymbol{x}_0 \sim p_0(\boldsymbol{x}), \boldsymbol{x}_T \sim p_T(\boldsymbol{x})} \left[\|(\boldsymbol{x}_T - \boldsymbol{x}_0) - \varphi(t, \boldsymbol{x}_t)\|^2\right] dt, \quad \boldsymbol{x}_t = t\boldsymbol{x}_T + (T - t)\boldsymbol{x}_0. \tag{2}$$

Here, the target distribution $p_T(\boldsymbol{x})$ and the initial distribution $p_0(\boldsymbol{x})$ are known. $\boldsymbol{x}_t$ for $t \in [0, T]$ is the interpolated point between $\boldsymbol{x}_T$ and $\boldsymbol{x}_0$, and the rectified flow encourages the vector field to align as closely as possible with the straight line $\boldsymbol{x}_T - \boldsymbol{x}_0$.

## 3 The Flow-Based Combinatorial Auction Menu Network

The major challenge in learning menus for CAs is to provide a suitable representation of bundle distributions. This representation should be efficient to avoid bottlenecks associated with the exponential number of bundles, while remaining differentiable to support gradient-based training.

### 3.1 Menu representation

Our key idea, inspired by score-based diffusion models and continuous normalizing flow, is to construct a concise and differentiable representation of a bundle distribution by modeling this through

the solution of an ODE. Specifically, the $k$th menu element generates its bundle distribution by the ODE,

$$d\boldsymbol{s}_t^{(k)} = \varphi(t, \boldsymbol{s}_t^{(k)})dt, \tag{3}$$

for a suitable vector field $\varphi$. The *bundle variable at time $t$*, $\boldsymbol{s}_t^{(k)} \in \mathbb{R}^m$, is generated from $\boldsymbol{s}_0^{(k)}$ by $\boldsymbol{s}_t^{(k)}(\boldsymbol{s}_0^{(k)}) = \boldsymbol{s}_0^{(k)} + \int_0^t \varphi(\tau, \boldsymbol{s}_\tau^{(k)})d\tau$. For clarity, we omit the superscript $(k)$ and the explicit dependence on $\boldsymbol{s}_0$ and write $\boldsymbol{s}_t$ when it is clear from the context.

We train the vector field $\varphi$ so that, at time $T$, a bundle variable $\boldsymbol{s}_T$ has all its entries being 0s or 1s and thus represents a valid bundle. By the Liouville equation [36], the allocation probability of this bundle, $\alpha_T(\boldsymbol{s}_T)$, derived from Eq. 3 satisfies:

$$\log \alpha_T(\boldsymbol{s}_T) = \log \alpha_0(\boldsymbol{s}_0) - \int_0^T \nabla \cdot \varphi(t, \boldsymbol{s}_t)dt, \tag{4}$$

where $\alpha_0(\boldsymbol{s}_0)$ is the initial probability and $\nabla \cdot \varphi(t, \boldsymbol{s}_t)$ is the divergence of $\varphi$.

**Training scheme**. Both the vector field $\varphi$ and the initial distribution $\alpha_0$ can influence the final distribution $\alpha_T$. Our method proceeds in two stages, involving the training of each of these two components in turn: *(1) Flow Initialization.* We fix the initial distribution $\alpha_0$ and train the vector field $\varphi(t, \cdot)$ so that the final distribution, $\alpha_T$, provides a reasonable coverage over bundles. *(2) Menu Optimization.* We fix the vector field from Stage 1, and backpropagate the revenue-maximizing loss through the flow to update the initial distribution $\alpha_0^{(k)}$ for each menu element $k$.

$\varphi$ and $\alpha_0(\boldsymbol{s}_0)$ play a crucial role in ensuring a concise and easily differentiable representation for efficient training. We next propose specific functional forms that meet these criteria.

**Vector field**. We adopt the following functional form for the vector field,

$$\varphi(t, \boldsymbol{s}_t; \xi, \theta) = \eta(t; \xi)Q(\boldsymbol{s}_0; \theta)\boldsymbol{s}_t, \tag{5}$$

where $Q : \mathbb{R}^m \to \mathbb{R}^{m \times m}$, written as a function of $\boldsymbol{s}_0$, and the scalar factor $\eta : \mathbb{R} \to \mathbb{R}$, written as a function of the ODE time $t \in [0, T]$, are neural networks with learnable parameters $\theta$ and $\xi$, respectively. We omit dependence on $\theta$ and $\xi$ when the context is clear. This formulation's advantage becomes apparent when we consider $\varphi$'s divergence:

$$\nabla \cdot \varphi(t, \boldsymbol{s}_t) = \sum_{i=1}^m \frac{\partial \varphi_i}{\partial s_{t,i}} = \sum_{i=1}^m \frac{\partial}{\partial s_{t,i}} \eta(t)Q_i(\boldsymbol{s}_0)\boldsymbol{s}_t = \sum_{i=1}^m \eta(t)Q_{ii}(\boldsymbol{s}_0) = \eta(t) \operatorname{Tr}[Q(\boldsymbol{s}_0)]. \tag{6}$$

Here, $\varphi_i$ and $s_{t,i}$ are the $i$th element of $\varphi$ and $\boldsymbol{s}_t$, respectively, $Q_i$ is the $i$th row of $Q$, and $Q_{ii}$ is the $i$th diagonal element of $Q$. Thus, the probability density at $T$ (Eq. 4) becomes

$$\log \alpha_T(\boldsymbol{s}_T) = \log \alpha_0(\boldsymbol{s}_0) - \operatorname{Tr}[Q(\boldsymbol{s}_0)] \int_0^T \eta(t)dt. \tag{7}$$

The integral in Eq. 7 is tractable as it only involves a scalar function, instead of bundle variables. We can efficiently estimate this integral by time discretization.

**Initial distribution**. In Stage 1, we use a mixture-of-Gaussian distribution to model $\alpha_0(\boldsymbol{s}_0)$, with $\boldsymbol{s}_0 \sim \sum_{d=1}^D w_d \mathcal{N}(\boldsymbol{\mu}_d, \sigma_d^2 \boldsymbol{I}_m)$, where, for $D$ components, $\boldsymbol{\mu}_d \in \mathbb{R}^m$, $\sigma_d \in \mathbb{R}_{>0}$, $\boldsymbol{I}_m$ is the $m \times m$ identity matrix, and $w_d \geq 0, \sum_{d=1}^D w_d = 1$. In Stage 2, as discussed later, we ensure DSIC by adopting a mixture-of-Dirac distribution for $\alpha_0(\boldsymbol{s}_0)$, which is operationalized by setting a very small variance $\sigma_d$ in the mixture-of-Gaussian distribution.

## 3.2 Stage 1: Flow initialization

Stage 1 ensures that the flow maps any initial bundle variable $\boldsymbol{s}_0$ to a feasible bundle $S \in 2^M$. Denote the indicator vector of $S$ by $\boldsymbol{s} = (\mathbb{I}\{i \in S\})$. $\boldsymbol{s}_i = 1$ if item $i$ is in $S$ and 0 otherwise. In practice, numerical issues make it challenging to obtain exactly $\boldsymbol{s}$. To account for this, we allow a small region around $\boldsymbol{s}$ to be approximated as $\boldsymbol{s}$ by modeling the bundle as a Gaussian variable, $S_{\sigma_z} = \mathcal{N}(\boldsymbol{s}, \sigma_z^2 \boldsymbol{I}_m)$. We thereby define the target distribution $\alpha_T^*(\boldsymbol{s}_T) = \frac{1}{2^m} \sum_{S \in 2^M} S_{\sigma_z}$ as a uniform mixture-of-Gaussian model, with components centered around each feasible bundle. This

target distribution only applies in Stage 1, where it serves to encourage a balanced coverage of the final distribution over feasible bundles. In Stage 2, we have an optimization problem, and there is no longer a fixed target distribution.

For Stage1, we fix the initial distribution $\alpha_0(s_0)$ to a mixture-of-Gaussian model $\alpha_0(s_0) = \sum_{d=1}^{D} w_d \mathcal{N}(\boldsymbol{\mu}_d, \sigma_d^2 \boldsymbol{I}_m)$ with $D$ components. The *flow training loss* is

$$\mathcal{L}_{\text{FLOW}}(\theta, \xi) = \mathbb{E}_{(s_0, s_T) \sim (\alpha_0, \alpha_T^*), t \sim [0,T]} \left[ \|(s_T - s_0) - \varphi(t, s_t; \theta, \xi)\|^2 \right], \tag{8}$$

$$s_t = t \cdot s_T + (T - t) \cdot s_0; \quad \varphi(t, s_t; \theta, \xi) = \eta(t; \xi) Q(s_0; \theta) s_t.$$

This loss is used to update the neural networks $Q$ and $\eta$ to encourage the vector field at interpolated points $s_t$ to point from $s_0$ to $s_T$. The expectation in the flow training loss is taken over $(\alpha_0, \alpha_T^*)$, but directly sampling from $\alpha_T$ is intractable as it involves $2^m$ bundles. Using a flow-based representation provides a workaround. We first draw $s_0 \sim \alpha_0$, which is straightforward given that $\alpha_0$ comprises a manageable number of components ($D$). We then round $s_0$ to the nearest feasible bundle, $s = \mathbb{I}(s_0 \geq 0.5) \in \{0, 1\}^m$, and sample $s_T \sim \mathcal{N}(s, \sigma_z^2 \boldsymbol{I}_m)$.

### 3.3 Stage 2: Menu optimization

In Stage 2, we train the menu to optimize the expected revenue. Trainable parameters are prices $\beta^{(k)}$, and $w_d^{(k)}$, $\boldsymbol{\mu}_d^{(k)}$ that define initial distributions $\alpha_0^{(k)}$. The vector field $\varphi$ is fixed in Stage 2 and shared among all menu elements. We always maintain a null menu element (zero allocation, zero price), which ensures individual rationality (IR), so that the bidder has non-negative expected utility.

Computing the expected utility corresponding to a menu element with bundle distribution $\alpha^{(k)}$ is central to evaluating the revenue objective but intractable when done with a direct calculation, because $u^{(k)}(v) = \sum_{S \in 2^M} \alpha^{(k)}(S) v(S)$ requires enumerating $2^m$ bundles for a general valuation function. By contrast, with a flow, we can get the bundle allocation probabilities by applying the exponential operation to both sides of Eq. 7 and writing

$$u^{(k)}(v) = \mathbb{E}_{s_0 \sim \alpha_0^{(k)}} \left[ \alpha_0^{(k)}(s_0) \exp \left( -\text{Tr}[Q(s_0)] \int_0^T \eta(t) dt \right) v \left( \lfloor s_T(s_0) \rceil \right) \right]. \tag{9}$$

Here, $\lfloor s_T(s_0) \rceil = \mathbb{I}(\phi(T, s_0) \geq 0.5)$ is the rounded final bundle, and $s_T(s_0)$ is the solution of the ODE, and solved by forward Euler,

$$s_T(s_0) = \phi(T, s_0) = s_0 + Q(s_0) \int_0^T \eta(t) s_t dt. \tag{10}$$

Due to its simple form, a modern ODE solver can efficiently solve the ODE (Eq. 10) in just a few steps. In this way, and since we also make the initial distribution simple, the calculation of $u^{(k)}(v)$ becomes tractable.

To ensure DSIC, we need to accurately calculate the expectation in Eq. 9 to get the exact utility to the bidder. We accomplish this by employing a mixture-of-Dirac distribution as the initial distribution, which has finite support. To implement this in practice, we set, for Stage 2 only, a very small variance to the Gaussian components, with $\sigma_d = 1e\text{-}20$ for every component $d$. In this way, the utility can be obtained by enumerating over the finite support of the initial distribution:

$$u^{(k)}(v) = \sum_{d=1}^{D} \left[ \alpha_0^{(k)}(\boldsymbol{\mu}_d^{(k)}) \exp \left( -\text{Tr}[Q(\boldsymbol{\mu}_d^{(k)})] \int_0^T \eta(t) dt \right) v(\lfloor s_T(\boldsymbol{\mu}_d^{(k)}) \rceil) \right], \tag{11}$$

where $\lfloor s_T(\boldsymbol{\mu}_d^{(k)}) \rceil = \mathbb{I}(\phi(T, \boldsymbol{\mu}_d^{(k)}) \geq 0.5)$. That is, the support of $\alpha_0^{(k)}$ consists, in effect, of the set of means, one for each component. It is worth noting that $D$ in Eq. 11 does not need to be the same $D$ as in Stage 1, and it could even vary across menu elements.

With $u^{(k)}(v)$ in Eq. 11, given a set of bidder valuations $\mathcal{V}$, the *revenue-maximization loss* is defined as

$$\mathcal{L}_{\text{REV}} \left( \{\beta^{(k)}\}_{k=1}^K, \{w_d^{(k)}\}_{\substack{d \in [D] \\ k \in [K]}}, \{\boldsymbol{\mu}_d^{(k)}\}_{\substack{d \in [D] \\ k \in [K]}} \right) = -\frac{1}{|\mathcal{V}|} \sum_{v \in \mathcal{V}} \left[ \sum_{k \in [K]} z^{(k)}(v) \beta^{(k)} \right], \tag{12}$$

Table 1: Revenue comparison on CATS. When $m = 10$, menus of baselines are large enough to accommodate all possible bundles. VCG has zero revenue for single-bidder CAs.

| Methods | CATS-Regions Uniform Valuations | | | | | CATS-Regions Normal Valuations | | | | |
|---|---|---|---|---|---|---|---|---|---|---|
| | $m$=10 | $m$=50 | $m$=75 | $m$=100 | $m$=150 | $m$=10 | $m$=50 | $m$=75 | $m$=100 | $m$=150 |
| BUNDLEFLOW | 196.19 | **555.05** | **454.96** | **417.83** | **385.68** | 173.93 | **603.70** | **448.35** | **389.51** | **394.82** |
| VCG | 0 | 0 | 0 | 0 | 0 | 0 | 0 | 0 | 0 | 0 |
| RochetNet | 189.17 | 288.41 | 290.14 | 292.11 | 312.65 | 167.16 | 270.23 | 300.55 | 270.23 | 291.83 |
| Grand Bundle | 162.57 | 316.27 | 321.10 | 317.00 | 314.93 | 142.54 | 319.06 | 328.89 | 305.06 | 309.73 |
| Menu+ | **202.26** | 399.85 | 354.48 | 322.68 | 329.17 | **181.93** | 459.97 | 405.36 | 306.60 | 312.21 |
| Menu- | 202.16 | 322.85 | 318.33 | 326.76 | 334.20 | 181.75 | 342.31 | 339.82 | 303.31 | 315.02 |

| Methods | CATS-Arbitrary Uniform Valuations | | | | | CATS-Arbitrary Normal Valuations | | | | |
|---|---|---|---|---|---|---|---|---|---|---|
| | $m$=10 | $m$=50 | $m$=75 | $m$=100 | $m$=150 | $m$=10 | $m$=50 | $m$=75 | $m$=100 | $m$=150 |
| BUNDLEFLOW | 211.55 | **560.71** | **467.79** | **434.77** | **420.75** | 235.80 | **646.37** | **490.03** | **428.54** | **394.37** |
| VCG | 0 | 0 | 0 | 0 | 0 | 0 | 0 | 0 | 0 | 0 |
| RochetNet | 205.02 | 316.79 | 312.41 | 313.76 | 334.51 | 221.49 | 348.41 | 312.79 | 304.00 | 316.81 |
| Grand Bundle | 175.09 | 329.62 | 340.40 | 343.66 | 345.98 | 186.59 | 354.86 | 345.88 | 329.99 | 336.45 |
| Menu+ | **233.26** | 396.78 | 351.70 | 357.44 | 351.07 | **248.16** | 478.67 | 349.34 | 335.75 | 339.10 |
| Menu- | 222.74 | 353.20 | 355.73 | 364.33 | 360.41 | 248.13 | 376.31 | 352.80 | 343.58 | 348.39 |

where $z^{(k)}(v) = \mathsf{SoftMax}_k \left( \lambda_{\text{SOFTMAX}} \cdot u^{(1)}(v), \ldots, \lambda_{\text{SOFTMAX}} \cdot u^{(K)}(v) \right)$ is a scaled SoftMax over bidder utilities and $\lambda_{\text{SOFTMAX}}$ is a scaling factor. In Stage 2, we fix the vector field $\varphi$ ($Q$ and $\eta$ networks) in Eq. 11 and update trainable parameters associated with the price and initial distribution $\alpha_0^{(k)}$, $i.e.$, $\beta^{(k)}$, $\{w_d^{(k)}\}_{d=1}^D$, and $\{\boldsymbol{\mu}_d^{(k)}\}_{d=1}^D$.

We formally prove that the learned mechanism is always DSIC.

**Theorem 1.** *[Exact DSIC] The* BUNDLEFLOW *framework ensures the learned auction mechanisms are DSIC.*

The detailed proof can be found in Appx. A.

## 4 Visualization on a Didactic Example

We present an example to visualize the BUNDLEFLOW training process, adopting the `Regions` environment from CATS [35] and considering a uniform value distribution and 5 items. For the flow network architecture, the $Q$ and $\sigma$ networks in this example each have three and two 128-dimensional, tanh-activated, fully-connected hidden layers, respectively. We use the Adam optimizer with a learning rate of $5e\text{-}3$ to train these networks, with 20K samples and 60K iterations. Fig. 1-left shows the effect of Stage 1, where we fix the initial distribution and update the vector field. The x- and y-axes are the bundle variables for the 3rd and 4th items, respectively ($x = 1$ and $y = 1$ mean that the 3rd and 4th items are in the bundle). The vector field gradually learns to cover all possible bundles in this projected, two-item subspace, as indicated by points $(0,0)$, $(0,1)$, $(1,0)$, and $(1,1)$ in the plots. Fig. 1-right illustrates the dynamics of Stage 2. The support size of the initial distributions is set to 512, the menu size to 8, and $\lambda_{\text{SOFTMAX}}$ to 1. The subplot is organized into four columns, each displaying changes in the bundle allocation (depicted by blue dots) and price of a distinct menu element. Changes in the bundle distribution for a menu element result from updates to the initial distribution for this element. We observe that each menu element learns to allocate different bundles. For example, Menu Element 3 manages bundles $(0,1)$ and $(1,0)$, and sets a price of 88.245 after Train Iteration 1020, at which point the revenue reaches 64.31.

## 5 Experiments

**Benchmark.** We evaluate our method on CATS [35], a standard benchmark in CA research.[1] Consistent with previous works [5, 6, 17, 19, 26–28, 47, 49, 59, 62], we focus our experiments on `Arbitrary` and `Regions` environments, which represent the most challenging problem instances [34]. Valuations are expressed in the CATS XOR bidding language as sets of bundles paired with their corresponding values (sets of atoms). We test different numbers of items: 10, 50, 75, 100, and 150

---

[1] We used the latest CATS v2.2 as is distributed under the "CATS License Agreement" (non-commercial research use); see `https://www.cs.ubc.ca/~kevinlb/CATS/`.

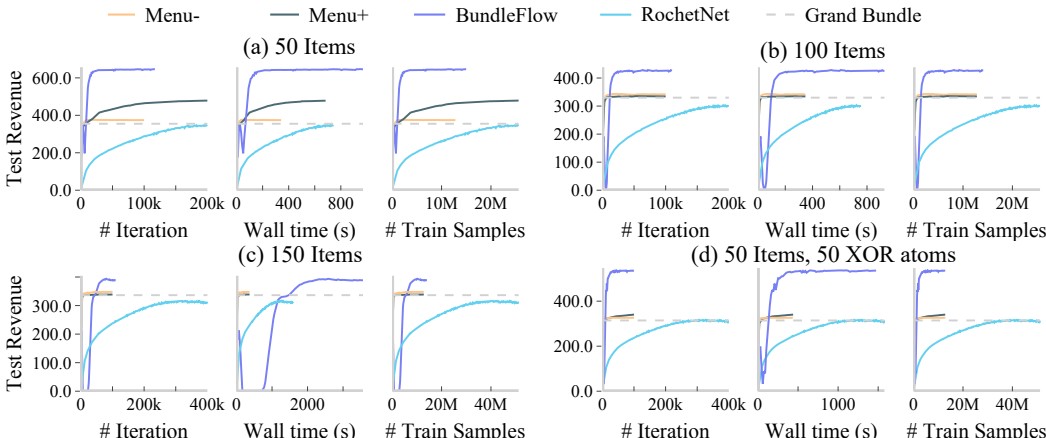

Figure 2: Learning curves on `CATS Arbitrary` with normal valuations. (a-c) Results for 50, 100, and 150 items, respectively. (d) Results for 50 items, increasing XOR atoms per value function from 5 to 50. Columns show the changes in test revenue as a function of the number of training iterations, wall time in seconds, and the number of training samples, respectively.

Table 2: Effect of the support size of the initial distribution ($D$) on BUNDLEFLOW in different CATS environments. The revenue drops dramatically when changing from $D = 2$ to $D = 1$.

| # Items ($m$) | CATS-Regions Normal Valuations | | | | | CATS-Arbitrary Uniform Valuations | | | | |
|---|---|---|---|---|---|---|---|---|---|---|
| | $D$=1 | $D$=2 | $D$=4 | $D$=8 | $D$=16 | $D$=1 | $D$=2 | $D$=4 | $D$=8 | $D$=16 |
| 50 | 278.82 | 589.50 | 596.67 | 603.70 | 595.14 | 287.98 | 557.06 | 563.70 | 560.71 | 561.51 |
| 100 | 258.18 | 364.29 | 372.49 | 389.51 | 388.06 | 279.75 | 425.94 | 426.80 | 434.77 | 428.53 |
| 150 | 291.69 | 338.52 | 368.00 | 394.28 | 383.24 | 318.97 | 383.44 | 396.71 | 420.75 | 411.86 |

across all environments. On the `Regions` environment with normal value distributions, we further test 200 and 500 items. When varying the number of items, we set the maximum XOR atoms per bid to 5 (the default value). We also experiment with increasing the maximum number of atoms to 50. Appx. D.1 discusses how to set up single-bidder auctions in CATS. All experiments are conducted on a single NVIDIA A100 GPU. Our code is available online[2].

**Baselines**. The classic Vickrey–Clarke–Groves (VCG) mechanism remains truthful in this single bidder CA context but charges the bidder her externality, which is zero when there is only one participant. Myerson's virtual-value approach [40] is designed for single-item auctions and is not well defined when items exhibit complementarities.

Learning-based methods that are DSIC fall in two broad categories. First, there are menu-based methods with item-wise allocations such as RochetNet [13]. These methods have limited power for CAs because they do not express general bundle distributions. We use RochetNet as the first baseline:

(1) `RochetNet` [13]: Interpret the item-wise allocation as a product distribution. Calculating bidder values remains intractable due to the need to enumerate all bundles. We address this by employing the *Gumbel-SoftMax* technique [31], which enables sampling from product distributions and backpropagation through the samples to update item-wise allocations.

Second, there are methods such as the *affine maximizer auction* (AMA) and its deterministic variant the VVCA [10–12]. These explore a restricted family of affine mechanisms, with the result that they exhibit restricted expressiveness and achieve suboptimal revenue. We include baselines that are simplified versions of the AMA and VVCA methods, where we necessarily fix the menu to contain a subset of all possible bundles in order to scale to instances with a large number of items:

(2) `Menu+`: The menu contains as elements the *grand bundle* (the bundle of all items), as well as the largest bundles immediately smaller in size. When the menu size cannot accommodate all bundles of a certain size, the selection of bundles of this size is random.

(3) `Menu-`: Similar to Menu+, but we begin by including elements that consist of single-item bundles, expanding as the menu size permits, and always including the grand bundle.

---

[2]`https://github.com/TonghanWang/BundleFlow.git`

Table 3: Effect of increasing $D$, the support size of initial distributions, under varying menu sizes $K$, the number of elements in a menu.

| Menu Size | | K/4 | | K/2 | | K | | K*2 | |
|---|---|---|---|---|---|---|---|---|---|
| **Environments** | $m$ | $D$=1 | $D$=2 | $D$=1 | $D$=2 | $D$=1 | $D$=2 | $D$=1 | $D$=2 |
| CATS-Regions | 50 | 228.34 | 575.52 | 259.95 | 568.94 | 278.82 | 589.50 | 315.88 | 602.98 |
| Normal | 100 | 213.26 | 324.04 | 226.11 | 345.51 | 258.18 | 364.29 | 270.65 | 419.12 |
| Valuations | 150 | 245.22 | 312.82 | 268.78 | 318.21 | 291.69 | 338.52 | 302.04 | 385.08 |
| CATS-Arbitrary | 50 | 238.96 | 546.50 | 261.61 | 551.78 | 287.98 | 557.06 | 303.53 | 561.28 |
| Uniform | 100 | 230.84 | 357.99 | 254.20 | 408.45 | 279.75 | 425.94 | 305.11 | 438.27 |
| Valuations | 150 | 297.22 | 355.07 | 303.08 | 361.01 | 318.97 | 383.44 | 343.19 | 404.30 |

(4) `Grand Bundle`: This menu has the grand bundle as the only menu element.

For (2) and (3), we learn the prices corresponding to each element using gradient-based optimization. For (4), the price on the single element, the grand bundle, is determined through a grid search to maximize training revenue. Performance, as with all methods, is reported on the test set.

**Hyperparameters**. Detailed hyperparameter settings of BUNDLEFLOW and baselines can be found in Appx. D. For BUNDLEFLOW, we set the support size of the initial distribution $D$ to 8, and the menu size $K$ to 5000 when $m \leq 100$, and 20000 otherwise. We did not extensively fine-tune the remaining hyperparameters, such as neural network sizes, as the initial trial guided by prior practice already yielded satisfactory results.

## 5.1 Experimental Results

**Performance**. Table 1 shows the revenue of our method and baselines. BUNDLEFLOW performs consistently and significantly better when the number of items is large ($10 < m < 200$), achieving significant revenue gains of $1.11-2.23\times$ over baselines. It remains superior at scale: for $200 \leq m < 500$, BUNDLEFLOW continues to outperform all the baselines as shown in Table 4. The learning curves in Fig. 2 reveal another advantage of BUNDLEFLOW. Compared to RochetNet, which is the baseline that also learns allocations in a menu, our method achieves a $3.6-9.5\times$ improvement in the number of training iterations in CATS `Arbitrary` with normal value distributions (for $10 < m < 200$). Our method can also reduce training time in some settings. For example, when $m = 50$, BUNDLEFLOW reduces the training duration from about 700 seconds to 140 seconds, achieving a $5\times$ improvement in training speed. These results highlight that BUNDLEFLOW, while allowing improved representational capacity over baselines, formulates an optimization problem that is tractable, enabling improved revenue as well as training efficiency. We discuss inference time in Appendix D.4.

When the number of items is small ($m$=10), baselines (`Menu+`, `Menu-`, `RochetNet`) with a menu size of 5K can already cover all possible bundles ($2^{10} = 1024$). Although these settings are not our primary focus, BUNDLEFLOW achieves comparable revenue to these baselines.

**The support size of bundle distributions**. Table 2 provides insights into why BUNDLEFLOW has superior revenue. When we reduce the support size of initial distributions ($D$) to 1, each menu element deterministically assigns a single bundle and no longer represents a bundle distribution. Correspondingly, we observe a dramatic drop in revenue when $D$ is decreased from $D = 2$ to $D = 1$. For example, in the CATS `Regions` with normal value distributions and 50 items, the revenue declines sharply from 589.50 to 278.82. This trend remains when we vary the menu size, as shown in Table 3. This pronounced performance gap between $D = 2$ and $D = 1$ highlights the importance of maintaining a randomized distribution over bundles in the application of differentiable economics to CAs—a capability uniquely enabled by our method.

In Table 5, we vary the menu size in a smaller range (10, 20, 100, 200) and present the test revenue of our method with $D = 1$, $D = 2$, and $D = 16$. When the menu size is small, increasing $D$ beyond 2 can triple the test revenue. With increased menu sizes, the gain for $D$ larger than 2 diminishes but remains positive. We interpret this observation as follows. The representation capacity depends on both the menu size and the support size of each menu element. For a small menu size, increasing $D$ beyond 2 continues to help. For a large menu size, the main gain comes from allowing some randomization (increasing $D$ from 1 to 2).

Table 4: Test revenue (large number of items).

| Methods | $m = 200$ | $m = 500$ |
|---|---|---|
| BUNDLEFLOW | **375.90** | **397.33** |
| Menu+ | 298.0 | 312.9 |
| Menu− | 302.8 | 315.0 |
| RochetNet | 365.0 | 375.0 |
| Grand Bundle | 296.3 | 308.5 |

Table 5: A support size of $D > 2$ contributes more significantly when the menu size is small.

| Menu Size | $D = 1$ | $D = 2$ | $D = 16$ |
|---|---|---|---|
| Menu Size 10 | 43.86 | 106.00 | 303.51 |
| Menu Size 20 | 67.84 | 272.12 | 337.82 |
| Menu Size 100 | 130.46 | 402.85 | 449.27 |
| Menu Size 200 | 148.47 | 439.61 | 449.18 |

**Menu size and valuation function size**. Table 3 presents the influence of menu sizes ($K$), specifically when they are halved, quartered, or doubled. A randomized distribution over bundles is crucial in each of these settings, with revenue falling sharply when $D$ decreases to 1. Furthermore, increasing the menu size tends to improve performance, although the gains are modest for a small number of items. For example, in CATS `Regions` with normal value distributions and $m = 100$, increasing the menu size from $K/4$ to $K \times 2$ leads to a 29.34% revenue boost, in contrast to a merely 4.77% boost when $m = 50$ . Increasing the maximum number of XOR atoms per valuation from 10 to 50 has minimal effects on our method and baselines, as evidenced in Table 6 and Fig. 3.

## 6 Closing Remarks

The key contribution in this paper is an ODE-based framework that enables concise yet flexible representation and efficient optimization for bundle distributions in CA deep learning. By keeping these distributions tractable—avoiding an exponential representation size—we dramatically increase revenue over baselines. Appx. B also suggests an interesting direction in extending this approach to multi-bidder CA design, although its current complexity is prohibitive. Reducing this complexity is key for future work.

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

# Appendix

## A   Proof

**Theorem 1.** *[Exact DSIC] The* BUNDLEFLOW *framework ensures the learned auction mechanisms are DSIC.*

*Proof.* The seminal work by Hammond [20] establishes necessary and sufficient conditions for a strategyproof menu-based auction: (1) Self-bid independent: the menu is independent of the bidder's bid; and (2) Agent-optimizing: the bidder is assigned the menu element that maximizes their utility. As we analyze here, our method satisfies these two properties.

In BUNDLEFLOW, all element prices, as well as bundle allocations, which depend on initial distributions and the vector field, are trained on values sampled from the distribution $F$, without using any information about the bidder's specific valuation. Therefore, menus learned by BUNDLEFLOW are self-bid independent.

As discussed in Sec. 3.3, we require the initial distribution for each menu element to have finite support, which means that the bundle distribution for each menu element can be reconstructed without any approximation error. This guarantees exact utility calculation for every menu element. Moreover, unlike the SoftMax during training, we use hard argmax at test time, thereby selecting the menu element with the highest utility to the bidder. In this way, BUNDLEFLOW is agent-optimizing. $\square$

## B   Discussion

**Representation flexibility**. In Stage 1, we initialize the vector field $\varphi$. The final distribution can in principle cover all $2^m$ bundles and is trained to seek to achieve this. In Stage 2, since the initial distribution for a menu element has finite support of size $D$, the bundle distribution for a menu element is also limited to finite support of size $D$. What is crucial, though, is that we can learn which (up to) $D$ bundles are represented in the distribution that corresponds to a menu element. In practice, we find that a bounded $D$ that is much smaller than $2^m$ still gives very high expected revenue.

**Extension to multi-bidder settings**. By providing a tractable and yet fully flexible representation of single-bidder menus for the CA setting, BUNDLEFLOW opens up the possibility of developing a general DSIC multi-bidder CA mechanism. A principled approach is to adapt the idea of GemNet [57]. First, we can learn a separate BUNDLEFLOW menu for each bidder. The modification in the network architecture is that these menus should now also depend on other bidders' bids $\boldsymbol{b}_{-i}$. To achieve this, we can condition the vector field, specifically the $Q$ and $\sigma$ networks, on $\boldsymbol{b}_{-i}$ by concatenating them to the inputs. For the price of each menu element, we can model them as the output of a neural network whose input is $\boldsymbol{b}_{-i}$. During training, we can also introduce a compatibility loss in the same way as that used in GemNet. This loss penalizes any over-allocation of items in the selected agent-optimizing elements from individual menus.

The major challenge in adapting GemNet to the multi-bidder CA setting will arise during the post-training stage of GemNet, which adjusts prices of menu elements so that there is provably never any over-allocation of items. For this, GemNet constructs a grid over the space of bidder values. On each grid point, GemNet formulates a mixed-integer linear program (MILP) to adjust prices to ensure that, the utility of the best element that is compatible with the choices of others in the sense of not over-allocating items is larger than that of all other elements by a safety margin. These safety margins prevent an incompatible menu element from being selected in the regions between grid points. Although the concise BUNDLEFLOW menu representation makes it possible for this MILP to be directly adapted to the CA setting and used to adjust BUNDLEFLOW menus to obtain a DSIC multi-bidder CA, the main issue is that the space of bidder values exhibits exponential dimensionality in the CA setting, resulting in an excessively large grid. Reducing this complexity represents the crucial remaining step in future work to enable the use of differentiable economics for learning a general, DSIC, and multi-bidder CA mechanism.

## C  Additional Background

**Bidding language**. In CAs, a suitable *bidding language* is critical to allow a bidder to report their bid without needing to enumerate a value for every possible bundle. There are many ways to do this (including text-based preference elicitation via an LLM [25, 52]), but a common approach is to use the *XOR bidding language*, which allows bidders to submit bid prices for each of multiple bundles under an exclusive-or condition; in effect, only one bid price on a bundle can be accepted. Popular CA testbeds such as CATS [35] and SATS [58] employ this bidding language extensively.[3] The semantics of the XOR bidding language is that the value on a bundle $S$ is the maximum bid price on any bundle $S'$, submitted as part of the XOR bid, and for which $S' \subseteq S$. XOR bids are succinct for valuation functions in which the bidder is only interested in a bounded number of possible bundles.

**Diffusion models**. Diffusion models have emerged as a powerful class of generative AI methods, spurring notable advances in a wide range of tasks such as image generation [14, 45], video generation [7, 21, 23], molecular design [18], text generation [39], and multi-agent learning [55]. At their core, these models perform a *forward noising process* in which noise is incrementally added to training data over multiple steps, gradually corrupting the original sample. A *reverse diffusion process* is then learned to iteratively remove noise, thereby reconstructing data from near-random initial states. In our setting, instead of reconstructing data, we extend the diffusion process to develop a tractable and differentiable method that optimizes a high-dimensional distribution.

In particular, *score-based diffusion models* enjoy strong mathematical and physical underpinnings. The forward noising process is an *Itô stochastic differential equation* (SDE),

$$dx = f(x, t)dt + h(t)dw, \tag{13}$$

where $x(t) \in \mathbb{R}^\ell$ is the *state* at time $t$, for some $\ell \in \mathbb{Z}_{>0}$, $f(\cdot, t) : \mathbb{R}^\ell \to \mathbb{R}^\ell$ is the *drift coefficient*, $h(\cdot) : \mathbb{R} \to \mathbb{R}$ is the *diffusion coefficient*, and $w$ is the *standard Wiener process* (Brownian motion). Different forward processes are designed by specifying functional forms for $f(\cdot, t)$ and $h(\cdot)$. The generation of data is then based on the reverse process, which is a diffusion process given by the *reverse-time SDE* [3],

$$dx = [f(x, t) - h(t)^2 \nabla_x \log q_t(x)]dt + h(t)d\bar{w}, \tag{14}$$

where $dt$ is an infinitesimal negative timestep, $\bar{w}$ is the *standard Brownian motion with reversed time flow*, and $q_t(x)$ *is the distribution of state* $x(t)$ *at time* $t$. The principal task in diffusion models is to learn the *score function*, $\nabla_x \log q_t(x)$, which has been effectively achieved using neural networks in recent work. This enables solving the reverse-time SDE and generating new data samples. Notably, in the diffusion model (and more broadly, generative AI) literature, $q_0(x)$ is typically a known target distribution over data samples from a pre-training dateset.

## D  Experiments

### D.1  Benchmark

In CATS, the valuation functions of bidders are recorded as bundle-bid pairs in an output file, with bundles from the same bidder identified by appending a dummy item tagged with a unique identifier. In effect, the bundle-bid pairs in an output file involving the same dummy item form an XOR representation of the bidder's valuation function. To obtain single-bidder valuations, we generate 100,000 such files and extract valuation functions identified by a consistent dummy item. Of these, 95% are used for training, with the remaining 5% reserved for testing.

### D.2  Hyperparameters

We do not extensively fine-tune hyperparameters. This suggests that our formulation of the ODE, including its functional form (Eq. 5) and initial conditions, is well-suited to the needs of the CA setting, making the optimization of the flow model relatively straightforward. Specifically, the $Q$

---

[3]When representing the values of multiple bidders these testbeds often also introduce so-called dummy items for distinguishing the bids of different bidders. Still, the semantics for a single bidder is, in effect, that of the XOR language.

Table 6: Revenue comparison against baselines across different CATS environments and value distributions. The number of items is fixed at $m = 50$, and we increase the valuation function size: $a = 10, 20, 30, 40$, and 50 as maximum XOR atoms per valuation, corresponding to the *maxbid* parameter in CATS.

| Environment | Baseline | $a = 10$ | $a = 20$ | $a = 30$ | $a = 40$ | $a = 50$ |
|---|---|---|---|---|---|---|
| CATS-Regions
Uniform Private Valuations | Grand Bundle | 314.46 | 309.11 | 316.04 | 320.61 | 314.35 |
| | Menu+ | 374.28 | 369.29 | 357.91 | 372.39 | 363.13 |
| | Menu- | 328.95 | 328.92 | 323.44 | 333.00 | 326.58 |
| | RochetNet | 308.81 | 313.24 | 317.14 | 318.15 | 310.60 |
| | BUNDLEFLOW | **533.42** | **528.46** | **528.26** | **539.92** | **537.36** |
| CATS-Regions
Normal Private Valuations | Grand Bundle | 322.74 | 301.54 | 310.38 | 304.38 | 334.95 |
| | Menu+ | 434.32 | 375.51 | 404.74 | 381.05 | 430.24 |
| | Menu- | 326.98 | 307.81 | 332.39 | 313.76 | 342.23 |
| | RochetNet | 301.47 | 297.34 | 317.10 | 302.04 | 327.32 |
| | BUNDLEFLOW | **564.13** | **524.24** | **535.35** | **525.68** | **566.51** |
| CATS-Arbitrary
Uniform Private Valuations | Grand Bundle | 331.21 | 334.43 | 330.33 | 351.01 | 336.68 |
| | Menu+ | 355.92 | 351.02 | 341.76 | 347.26 | 348.92 |
| | Menu- | 352.34 | 354.58 | 346.36 | 354.87 | 353.58 |
| | RochetNet | 329.61 | 345.06 | 338.44 | 344.76 | 350.75 |
| | BUNDLEFLOW | **565.54** | **583.60** | **568.01** | **579.01** | **581.39** |
| CATS-Arbitrary
Normal Private Valuations | Grand Bundle | 381.51 | 335.44 | 323.52 | 320.28 | 358.83 |
| | Menu+ | 470.63 | 349.70 | 346.39 | 340.59 | 381.13 |
| | Menu- | 402.23 | 341.79 | 337.44 | 332.90 | 368.26 |
| | RochetNet | 367.44 | 329.28 | 333.88 | 327.57 | 365.40 |
| | BUNDLEFLOW | **664.94** | **564.90** | **552.77** | **548.22** | **615.93** |

network comprises three 128-dimensional tanh-activated fully connected layers. When $m > 100$, we increase the width of the last layer to 256. The $\sigma$ network is simpler and has two 128-dimensional tanh-activated fully connected layers.

Two important hyperparameters are $D$, the support size of the initial distribution, and $K$, the menu size. By default, $D$ is set to 8, a relatively small number. $K$ is 5000 when $m \leq 100$ and is 20000 otherwise. The same menu size is used for our method, Menu-, Menu+, and RochetNet. Notably, the menu size $K$ is adequate to encompass all possible bundles for smaller numbers of items, such as $m = 5$ or 10. We show the impact of different values of $D$ and $K$ in ablation studies.

Menu optimization for BUNDLEFLOW is conducted using the Adam optimizer with a learning rate of 0.3. $\lambda_{\text{SOFTMAX}}$ is increased from 0.001 to 0.2 over the course of training. For comprehensive details on our hyperparameter settings, please refer to the codebase. For the baselines, we fine-tuned their hyperparameters so that they perform significantly better than the default RochetNet setting. The modifications are achieved by performing a grid search to obtain the optimum combination of $\lambda_{\text{SOFTMAX}}$ and learning rate that yields the best revenue and also guarantees convergence. Both Menu- and Menu+ use a learning rate of 0.3 and $\lambda_{\text{SOFTMAX}}$ of 2, while RochetNet uses a learning rate of 0.05 and $\lambda_{\text{SOFTMAX}}$ of 20.

All experiments are conducted on a single NVIDIA A100 GPU.

### D.3 Baselines.

We give an example of allocations in the menu of Menu+. With 3 items and a menu size of 3, the menu would include $[1, 1, 1]$ (the grand bundle) and a random subset of bundles containing 2 items, such as $[1, 1, 0]$ and $[0, 1, 1]$.

### D.4 Inference speed.

Inference for our method is fast in absolute time: for $m = 100$ with a batch size of 200 samples, BUNDLEFLOW runs in $4.74 \times 10^{-3} \pm 2.27 \times 10^{-4}$ s, compared with RochetNet at $6.39 \times 10^{-4} \pm$

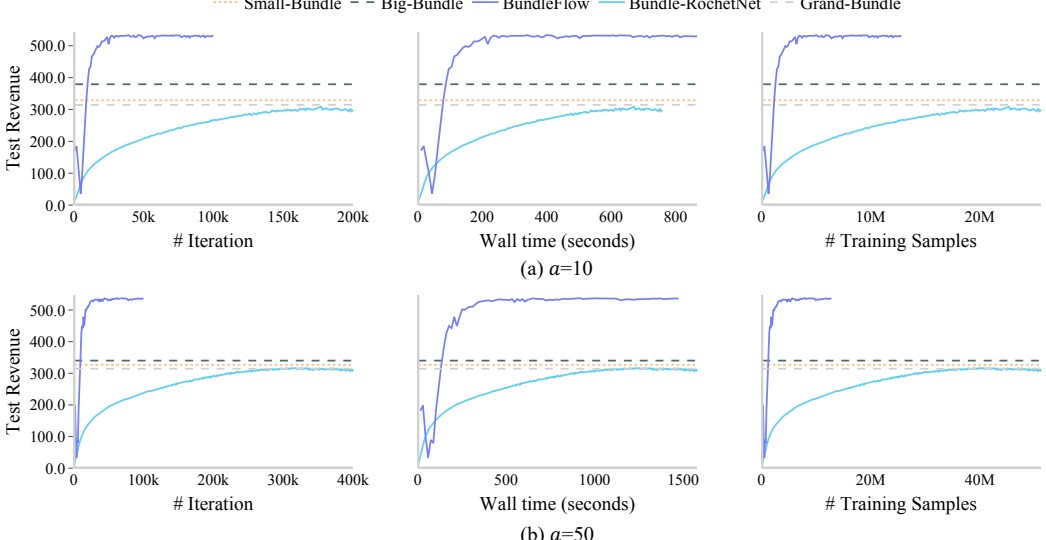

Figure 3: Learning curves of BUNDLEFLOW and baselines on `CATS Regions` with uniform valuation distributions with different valuation function sizes. The two rows are results for $a = 10$ and $50$ XOR atoms per valuation, respectively. The three columns show the changes of test revenue as a function of the number of training iterations, wall time in seconds, and the number of training samples, respectively.

$3.01 \times 10^{-5}$ s, `Menu+` at $3.70 \times 10^{-5} \pm 4.70 \times 10^{-6}$ s, and `Menu-` at $3.60 \times 10^{-5} \pm 2.30 \times 10^{-6}$ s under the same conditions. While our method has a relatively higher inference time, it requires significantly fewer training iterations to converge, maintaining a clear advantage in terms of overall training time. After training, our method requires deploying a menu, where the allocation and price of each menu element can be calculated before deployment. Users only need to check which bundles (whose number is upper bounded by a constant) are in a menu element, calculate its utility, and assign the best menu element.

