# OpenReview forum: "BundleFlow: Deep Menus for Combinatorial Auctions by Diffusion-Based Optimization"
_NeurIPS.cc/2025/Conference — NeurIPS 2025 poster_

### Official Review · Reviewer_kwyj · 2025-06-25

**Clarity:** 2
**Significance:** 2
**Originality:** 2
**Rating:** 4
**Confidence:** 2

**Summary:**

This paper attempts to solve the combinatorial auction problem and proposes bundflow, a menu-based mechanism that represents the bundle distribution of each menu option via a continuous normalizing flow. This method avoids the exponential blow-up in explicit bundle representations and preserves exact dominant-strategy incentive compatible. Experiments and theoretical analysis are conducted.

**Questions:**

Please see the weaknesses section.

Based on the current presentation and considering the novelty, I tend to recommend acceptance. However, I'm not an expert in this domain and am open to revising my decision according to future discussions.

**Ethical Concerns:**

["NO or VERY MINOR ethics concerns only"]

**Final Justification:**

During the rebuttal stage, my concerns regarding presentation improvements and ablation studies have been largely addressed. While I maintain some reservations about the presentation improvements, since the authors have only committed to enhancing them rather than demonstrating concrete changes. After thoroughly reviewing all reviewer comments and author responses, I believe this is not a significant concern to change my rating. Therefore, I choose to maintain my original rating and vote for acceptance.

**Quality:**

2

**Strengths And Weaknesses:**

**Strengths**

1. Using the normalizing flow to solve the combinatorial auction problem seems new in the literature.

2. The sample codes are provided to help the reviewer verify independently.


**Weaknesses**

1. The presentation of this paper could be improved. For example, the proof of Theorem 1 is not self-contained. Based on the current version, Theorem 1 seems to be a simple extension/corollary of results in [17].

2. It would be better to include the wall-clock training and inference times (and GPU type) for all baselines to support the 80 % saving claim clearly.

---

> ### Author Rebuttal · Authors · 2025-07-31
>
> Thank you for your valuable comments.
>
> > Question 1: `The presentation of this paper could be improved. For example, the proof of Theorem 1 is not self-contained. Based on the current version, Theorem 1 seems to be a simple extension/corollary of results in [17].`
>
> We will reorganize the proof argument of Theorem 1 in the main body of the paper. The crucial part is the discussion presented between Lines 226 and 239, which builds upon Eqs. 11 and 12. The analysis surrounding Eqs. 11 and 12 is used to prove that our method can guarantee the bidder-optimizing condition, thereby establishing strategy-proofness.
>
> The technical conditions required for strategy-proofness and IR of single-bidder auctions are standard. What is challenging in our setting is that it must be tractable to compute the best menu option (at inference time) to fulfill the bidder-optimizing condition. A menu element can, in theory, involve a randomized allocation over an exponential number of bundles. This is one motivation for us to introduce ODEs to represent the bundle distribution.
>
> > Question 2: `It would be better to include the wall-clock training and inference times (and GPU type) for all baselines to support the 80 % saving claim clearly.`
>
> The only baseline for which we do not report wall-clock training times in Figures 2 and 3 is *Grand Bundle*. The reason is that *Grand Bundle*  is not a deep learning method. It performs a grid search over a single price for a fixed bundle that contains all items. When claiming the speed advantage of our method in the main paper, we made it clear that we are comparing with other *menu-learning* approaches, and the reported 80% reduction refers specifically to training time.
>
> In Appendix D.2 we reported that all experiments, including our method and the baselines, are conducted on a single NVIDIA A100 GPU. We will move this detail to the main text to improve clarity.
>
> As for the inference time, for 100 items and a batch of 200 samples, our method takes $0.00474 \pm 0.000227$ seconds per batch on an A100 GPU. For comparison, the inference time of *RochetNet*, *Menu+*, and *Menu-* under the same conditions is $0.000639  \pm 3.01e-05$, $3.7e-05\pm 4.7e-06$, $3.6e-05 \pm 2.3e-06$ seconds, respectively. We will add this inference time information to the paper. While our method has a higher inference time, it requires significantly fewer training iterations to converge, maintaining a clear advantage in terms of overall training time. After training, our method requires deploying a menu, where the allocation and price of each menu element can be calculated before deployment. Users only need to check which bundles (whose number is upper bounded by a constant) are in a menu element, calculate its utility, and assign the best menu element.

---

> > ### Comment · Reviewer_kwyj · 2025-08-05
> >
> > Thanks for the authors' detailed response. I've read them as well as the comments from other reviewers. Some of my concerns are addressed. I will keep my current rating.

---

> > > ### Author Response · Authors · 2025-08-06
> > >
> > > We sincerely thank the reviewer for the time, thoughtful comments, and valuable feedback. We will improve the proof of Theorem 1 to enhance its clarity, and we will provide additional details on wall-clock training and inference times in the next version. We would be greatly appreciative if the reviewer can let us know if there are any remaining questions. We would be grateful for the opportunity to address them.

---

> > > ### Author Response · Authors · 2025-08-08
> > >
> > > Dear Reviewer kwyj,
> > >
> > > Thank you once again for your thoughtful and constructive feedback. We’re encouraged to hear that some of your concerns have been addressed. We would be grateful if you could let us know if there are any remaining questions or issues we should address. If additional technical details, experimental clarifications, presentation refinements, etc., would help, we would be happy to provide them.
> > >
> > > We deeply appreciate the time and care you have devoted to reviewing our work.
> > >
> > >
> > >
> > >  &nbsp;
> > >
> > > Warm regards, \
> > > Authors of Submission 12959

---

> ### Comment · Area_Chair_BfyS · 2025-08-05
>
> This is a gentle reminder that the authors have responded to your question. Could you please provide your feedback on their response?
>
> Kindly note that the deadline of August 6 is approaching, and your timely feedback would be greatly appreciated to facilitate any further discussions if needed.

---

### Official Review · Reviewer_TcJ8 · 2025-06-28

**Clarity:** 3
**Significance:** 3
**Originality:** 3
**Rating:** 5
**Confidence:** 3

**Summary:**

The paper deals with single-bidder combinatorial auctions. The paper proposes a new method inspired by diffusion which consists of two stages. In the first stage a diffusion model is trained so that the entire space of bundles is in its support. The second stage uses this distribution for optimization by manipulating the initial distributions in order to maximize the revenue.

**Questions:**

-

**Ethical Concerns:**

["NO or VERY MINOR ethics concerns only"]

**Final Justification:**

The authors have answered my questions and I am maintaining my score.

**Limitations:**

yes

**Paper Formatting Concerns:**

-

**Quality:**

3

**Strengths And Weaknesses:**

### Strengths:
- Well written and clearly motivates the single bidder setting of combinatorial auctions whereas most methods I am familiar with only discuss the multi-bidder case. Figure 1 is very didactic and explains the method beautifully.
- Strong empirical results, clearly demonstrating the method's superior performance compared to baselines.

### Weaknesses:
- Can you clarify why your method only scales to 150 items? Do you see a degradation in performance or does it become too computationally expensive?
- Going from D=1 to D=2 seems lead to most of the performance improvement which I didn't fully understand. Can you clarify why increasing D further only provides marginal benefits?

---

> ### Author Rebuttal · Authors · 2025-07-31
>
> Thank you for your valuable comments.
>
> > Question 1: `Can you clarify why your method only scales to 150 items? Do you see a degradation in performance or does it become too computationally expensive?`
>
> Although the main paper reports experiments up to 150 items, our method scales further. In the tables below, we show the test revenue on the CATS regions environment with normal value distributions, for 200 and 500 items, respectively.
>
> **Table 1: With 200 items, compare the test revenue of our method against baselines.**
>
> | Menu+ | Menu- | RochetNet | Grand Bundle | BundleFlow (Ours) |
> | --- | --- | --- | --- | --- |
> | 298.0 | 302.8 | 365.0 | 296.3 | **375.90** |
>
> **Table 2: With 500 items, compare the test revenue of our method against baselines.**
>
> | Menu+ | Menu- | RochetNet | Grand Bundle | BundleFlow (Ours) |
> | --- | --- | --- | --- | --- |
> | 312.9 | 315.0 | 375.0 | 308.5 | **397.33** |
>
> Our method continues to outperform all baselines. Despite this success, a challenge that arises for larger numbers such as 500 items is with the first training stage, where we seek to initialize the flow so that the induced distribution covers all possible bundles. Empirically, we find through sampling that the induced distribution becomes less uniform and more skewed towards larger bundles. We conjecture that this phenomenon is attributable to the representational capacity of the flow network (and can be addressed in future work through increased capacity).
>
> > Question 2: `Going from D=1 to D=2 seems lead to most of the performance improvement which I didn't fully understand. Can you clarify why increasing D further only provides marginal benefits?`
>
> - **Why $D=2$ makes a big difference compared to $D=1$**. When $D=1$, each menu
> element must deterministically assign a single bundle. $D=2$ is the smallest possible value that allows a randomized bundle distribution. This is a discontinuity, and a qualitative change.
> - **Behavior for $D>2$ depends on the menu size**. In Table 3, we show the results of a new experiment, where we vary the menu size in a lower range (10, 20, 100, 200) and present the test revenue of our method with $D=1$,  $D=2$, and  $D=16$.
>
> **Table 3: A support size of $D>2$ contributes more significantly with a small menu size.**
>
> | Menu Size | D=1 | D=2 | D=16 |
> | --- | --- | --- | --- |
> | Menu Size 10 | 43.86 | 106.00 | 303.51 |
> | Menu Size 20 | 67.84 | 272.12 | 337.82 |
> | Menu Size 100 | 130.46 | 402.85 | 449.27 |
> | Menu Size 200 | 148.47 | 439.61 | 449.18 |
>
> *Observations:* When the menu size is small, increasing $D$ beyond 2 can *triple* the test revenue. With increased menu sizes, the gain from $D>2$ diminishes but remains positive.
>
> *Analysis:* We interpret this new observation as follows. The representation capacity, in regard to menus, depends on both the menu size and the support size of each menu option. For a small menu size, increasing $D$ beyond 2 continues to help. For a large menu size, the main gain comes from allowing some randomization (increasing  $D$ from 1 to 2).

---

> ### Author Response · Authors · 2025-08-06
>
> We sincerely thank the reviewer for the encouraging feedback and thoughtful suggestions. We greatly appreciate the insights provided, which will help improve the paper. We will incorporate the new experimental results in the next version.

---

### Official Review · Reviewer_iwrK · 2025-07-01

**Clarity:** 4
**Significance:** 2
**Originality:** 4
**Rating:** 4
**Confidence:** 4

**Summary:**

This paper studies menu-based auctions for single-bidder combinatorial auctions. The authors proposed a diffusion-based computational techniques for generating menus. Experimental results show that the technique scales really well. It can handle 150 items with ease (less than 1 hour). The achieved revenue is significantly better a few baselines.

**Questions:**

Can you give some practical scenarios where we need to solve single-bidder CA auction for 50-150 items? I am happy to change my rating if I see a good practical scenario. As I mentioned earlier, I think the techniques used in this paper are of high quality and original.

**Ethical Concerns:**

["NO or VERY MINOR ethics concerns only"]

**Final Justification:**

The authors' rebuttal contains very convincing examples on the practicality of the studied model.

**Limitations:**

This paper studies a theoretical auction model. The data are synthetic. I don't see any potential negative societal impact.

**Paper Formatting Concerns:**

No formatting concerns.

**Quality:**

4

**Strengths And Weaknesses:**

Since the authors focused on single-bidder menu-based auctions. Strategy-proofness and individual rationality are essentially non issues. As long as the menus are presented to the bidder (with the option of not paying and get nothing), and the bidder picks the best menu option, then SP and IR are achieved. Even when restricted to a single bidder, the setting is well motivated and has significance in practise.

The authors presented the main computational challenge via an example. What does [0.3, 0.6] mean for two items? As these can be achieved via different ways to group the deterministic bundles. The main compuational challenge is there are exponential number of ways to achieve a given item vector. To address this, the authors' main technique is to use an ODE to describe how the bundle breakdown is generated, and this ODE is characterized via a vector field. The first step of the proposed computational technique is to learn this vector field so that it generates feasible determinsitic bundle breakdowns.  The second step is to maximize revenue via menu optimization. Again this step is technically sophisticated, relying on ODE manuveour for the actual training.

The main downside of the proposed approach is its practical significance. Technically, the proposed technique makes a lot of sense. It is sophisticated and it does what is designed to do well, i.e., proposing good solutions for *very large* number of items. When we have very large number of items, we cannot use AMA/VVCA (and other heuristic methods for revenue maximization), the authors used baselines like menu+, menu-, grand, rochetnet, which are designed for scalability. The proposed method convincingly beat all existing baselines for large number of items. However, when the number of items is small, say, 10 items, the proposed method does not perform better than all baselines, as shown in Table 1. I totally understand that for this diffusion-based method, its advantage is to handle large number of items as it handles scalability really well. My main concern is whether single-bidder 50 item CA is an important or even a practically realistic setting. We are talking about 2^50 bundles. What is a realistic situation for this kind of scale? The main contribution of this paper is a technique to push up scalability. My question is whether scalability is actually an issue for the studied model.

---

> ### Author Rebuttal · Authors · 2025-07-31
>
> Thank you for the valuable comments.
>
> > Main question:  `The main downside of the proposed approach is its practical significance. Can you give some practical scenarios where we need to solve single-bidder CA auction for 50-150 items?`
>
> 1. Many examples of single-bidder CA problems involve a large number of items. Here we give three real-world examples with references.
>
>     - ___(Scenario 1) Selling on-demand cloud compute___. AWS [1] is a typical example. In this setting, an item is characterized by the number of GPUs, CPUs, Instance Memory (GiB), GPU Memory (GiB), Network Bandwidth, PUDirect RDMA, Instance Storage (TB), EBS Bandwidth (Gbps), etc. This yields hundreds of items. AWS already lists 850+ instances (”pre-defined bundles”) for users to choose from.  Single-bidder CAs can be used to design and price the instances.
>     - ___(Scenario 2) Selling a bundle of information goods___, e.g., selected from artists or TV shows or podcasts. As an example, on March 02, 2022, Prime Video UK [2] offered 65+ channels (such as Discovery, ITV Hub+, Eurosport Player, hayu, STARZPLAY and more). Users can pay for the ones they want. Optimally pricing and configuring each of the different subsets is an example CA of the size analyzed in this paper.
>     - ___(Scenario 3) Selling configurable managed services___. This is quite common in daily life, for example, a building operations service, where choices involve weekly frequency of trash pickup for different types of trash, different types and sizes of security patrols, method and frequency of cleaning of floors, different types of printers, and different types of A/V IT support. As an instance, Aramark [3] provides at least 9 service lines (ops & maintenance, custodial, engineering, HVAC, electrical, pest control, building intelligence, staffing, energy). Selecting a service level (e.g., daily, weekly, monthly) for each line across 15 buildings easily yields hundreds of items.
>
> 2. Consider the next step of research, where the goal will be to extend our method to multi-bidder cases. Wireless spectrum rights auctions, run by governments around the world, handle 3,000-12,000 licenses.
>
> 3. We have also completed additional experiments to evaluate the usefulness of our method on smaller problems. Although our method does not surpass baselines in 10-item CAs, these results show that it starts to outperform all baselines when the number of items increases to 12.
>
> [1] https://aws.amazon.com/ec2/instance-types
>
> [2] https://press.aboutamazon.com/uk/2022/3/prime-video-pledges-10m-to-support-training-and-development-in-the-uk-tv-and-film-industry
>
> [3] https://www.aramark.com/our-services/facilities-management-services
>
> > Comment 1: `Strategy-proofness and individual rationality are essentially non issues.  As long as the menus are presented to the bidder (with the option of not paying and get nothing), and the bidder picks the best menu option.`
>
> The technical conditions required for strategy-proofness and IR of single-bidder auctions are standard. What is challenging in our setting is that it must be tractable to compute the best menu option (at inference time). A menu element can, in theory, involve a randomized allocation over an exponential number of bundles. This is also one motivation for us to introduce ODEs to represent the bundle distribution.
>
> > Comment 2: `What does [0.3, 0.6] mean for two items? `
>
> This is an item-wise allocation, meaning that the bidder has a probability of 0.3 and 0.6 of getting item 1 and item 2, respectively. As the reviewer pointed out, this item-wise allocation can be achieved in different ways of grouping deterministic bundles.

---

> > ### Comment · Reviewer_iwrK · 2025-08-05
> >
> > I think the provided practical scenarios are very convincing.
> >
> > I have raised my score.

---

> ### Author Response · Authors · 2025-08-06
>
> We sincerely thank the reviewer for the thoughtful and constructive questions and feedback, which will help enhance the quality of the paper. We will incorporate this discussion on practical scenarios for single-bidder combinatorial auctions involving 50–150 items into the next version.

---

### Official Review · Reviewer_B4jQ · 2025-07-06

**Clarity:** 3
**Significance:** 3
**Originality:** 3
**Rating:** 5
**Confidence:** 1

**Summary:**

This paper presents BUNDLEFLOW, the first DSIC and revenue-optimizing deep learning method for single-bidder combinatorial auctions (CAs). It overcomes the exponential bundle space challenge by generating bundle distributions using ODE-based transforms, one per menu element. The method significantly outperforms baselines in revenue (up to 2.23×), reduces training iterations by up to 9.5×, and cuts training time by ~80% for large item sets (e.g., 50–150 items).

**Questions:**

NA

**Ethical Concerns:**

["NO or VERY MINOR ethics concerns only"]

**Final Justification:**

I maintain my score for Accept.

The authors are able to provide sound details regarding technical considerations and a commitment to including the recommended parameters in the source code, which will support the technical progress in the area.

**Limitations:**

yes

**Paper Formatting Concerns:**

No issue is identified regarding the submitted paper content, paper references, the NeurIPS paper checklist.

**Quality:**

3

**Strengths And Weaknesses:**

Strength:
1. Scalable and efficient: Achieves superior performance and faster training in large-scale CA settings.

2. Novel technique: Introduces an ODE-based randomized menu approach that balances flexibility and tractability.

Weakness:
1. The usage of ODE training and hyperparameter tuning may pose a barrier for practical deployment.

---

> ### Author Rebuttal · Authors · 2025-07-31
>
> Thank you for your valuable comments.
>
> > Question: `The usage of ODE training and hyperparameter tuning may pose a barrier for practical deployment.`
> >
>
> Thanks for the question. We provide a detailed analysis of which parts might be barriers to practical deployment and make recommendations about how to mitigate barriers.
>
> **1. The impact of ODE training on practical deployment**
>
> - __Compact closed form of the ODE__. We understand that an ODE might raise the concern because solving it or calculating the final probability distribution involves an integral. To deal with this issue, we propose a special functional form for the ODE (Eq. 5), which simplifies the integral so that it involves only a scalar function. Such a formulation enables fast inference.
> - __Training and deployment in practice__. The lighter ODE computation contributes to the training speed. Compared to RochetNet, which is a baseline that also trains both allocation and prices in a menu of the same size, our method reduces the training duration from about 700 seconds to 140 seconds on the same GPU. After training, our method requires deploying a menu, where the allocation and price of each menu element can be calculated before deployment. Users only need to check which bundles (whose number is upper bounded by a constant $D$) are in a menu element, calculate its utility, and assign the best menu element.
>
> **2. The impact of hyperparameter tuning on practical deployment**
>
> - __Robustness to some hyperparameters__. We find that our method is robust to some hyperparameters, e.g., the neural network architecture, the batch size for menu optimization, and those regarding the initial distributions. For these design choices, the default hyperparameter settings in our codebase are a sensible choice for accelerating the development process.
> - __Recommendations for the others__. There are other hyperparameters that need fine-tuning. During our experiments, we find that the learning performance is sensitive to softmax temperature $\lambda_{\rm SoftMax}$ and the learning rate for menu optimization. As detailed in Appendix D.2, we provide generally effective configurations. The published code will include recommended settings for these hyperparameters tailored to CA problems of different scales, enabling users to select appropriate values based on their specific applications.

---

> > ### Comment · Reviewer_B4jQ · 2025-08-03
> >
> > I thank the authors for providing a candid and detailed response. I appreciate the technical considerations and the author's commitment to including the recommended parameters in the source code, which will support the technical progress in the area. I maintain my score for Accept.

---

> ### Author Response · Authors · 2025-08-06
>
> We sincerely thank the reviewer for the encouraging feedback and valuable insights, which will help strengthen the paper! We will incorporate the recommended hyperparameter settings in the source code.

---

### Decision · Program_Chairs · 2025-09-17

**Decision:**

Accept (poster)

**Comment:**

This paper introduces a DSIC and revenue-optimizing deep learning approach for single-bidder combinatorial auctions (CAs). The core challenge addressed is the exponential bundle space, which has traditionally made scalable mechanism design in this setting intractable. It leverages an ODE-based continuous transform to generate bundle distributions for each menu element, enabling both tractability and expressivity.

The paper makes a novel and technically rigorous contribution to the design of revenue-optimal, DSIC mechanisms in combinatorial auctions. Despite some concerns about practical applicability and presentation (which are partially resolved during rebuttals), the scalability results, theoretical grounding, and methodological innovation clearly outweigh the weaknesses.